# Incidence of dengue illness in Mexican people aged 6 months to 50 years old: A prospective cohort study conducted in Jalisco

Rodrigo DeAntonio[1¤], Gerardo Amaya-Tapia[2], Gabriela Ibarra-Nieto[2], Gloria Huerta[3], Silvia Damaso[4], Adrienne Guignard[4], Melanie de Boer[5]*

1 Vaccines, GSK, Panama City, Panama, 2 Department of Infectious Diseases, Hospital General de Occidente, Zapopan, Mexico, 3 Vaccines, GSK, Mexico City, Mexico, 4 Vaccines, GSK, Wavre, Belgium, 5 Vaccines, GSK, Rockville, Maryland, United States of America

¤ Current address: Centro de Vacunación Internacional (Cevaxin), Panama City, Panama
* melanie.x.de-boer@gsk.com

## Abstract

### Background and objectives

The burden of dengue virus (DENV), a mosquito-borne pathogen, remains difficult to assess due to misdiagnosis and underreporting. Moreover, the large proportion of asymptomatic dengue cases impairs comprehensive assessment of its epidemiology even where effective surveillance systems are in place. We conducted a prospective community-based study to assess the incidence of symptomatic dengue cases in Zapopan and neighboring municipalities in the state of Jalisco, Mexico.

### Methods

Healthy subjects aged 6 months to 50 years living in households located in the Zapopan and neighboring municipalities were enrolled for a 24-month follow-up study (NCT02766088). Serostatus was determined at enrolment and weekly contacts were conducted via phone calls and home visits. Participants had to report any febrile episode lasting for at least two days. Suspected dengue cases were tested by reverse-transcriptase quantitative polymerase chain reaction (RT-qPCR), detection of non-structural protein 1 (NS1), anti-DENV immunoglobulin G and M (IgG and IgM) assays.

### Results

A total of 350 individuals from 87 households were enrolled. The overall seroprevalence of anti-DENV IgG at enrolment was 19.4% (95% confidence interval [CI] 14.5–25.6) with the highest seroprevalence rate observed in the adult group. Over the 27-month study period from July 2016 to September 2018, a total of 18 suspected dengue cases were reported. Four cases were confirmed by RT-qPCR and serotyped as DENV-1. A fifth case was confirmed by the NS1 assay. The 13 remaining suspected cases were tested negative by these assays. Based on the 5 virologically confirmed cases, symptomatic dengue incidence

**Data Availability Statement:** For reasons of privacy protection for study participants, GSK offers access to data and materials via controlled

access. Anonymized individual participant data from this study plus the annotated case report form, protocol, reporting and analysis plan, data set specifications, raw dataset, analysis-ready dataset, and clinical study report are available for research proposals approved by an independent review committee. Proposals should be submitted to www.clinicalstudydatarequest.com (study identifier: 200318). A data access agreement will be required. The authors did not have any special privileges in accessing the data that other researchers would not have.

**Funding:** GlaxoSmithKline Biologicals SA funded this study (NCT02766088/GSK study identifier: 200318). GlaxoSmithKline Biologicals SA also provided support in the form of salaries for authors GH, SD, MDB and AG. GlaxoSmithKline Biologicals SA was involved in all stages of study conduct, including study design, data collection and analysis; GlaxoSmithKline Biologicals SA also covered all costs associated with the development and publication of this manuscript.

**Competing interests:** GlaxoSmithKline Biologicals SA (GSK) funded this study and covered all costs associated with the development and publication of this manuscript. GH, SD, MDB and AG are employees of the GSK group of companies. SD and AG hold shares in the GSK group of companies. RD was an employee of the GSK group of companies at the time of the study. GAT received payments from the GSK group of companies, as part of the multi-center study. GH, SD, MDB, AG and GAT declare no other financial and nonfinancial relationships and activities. GIN declare no financial and non-financial relationships and activities and no conflicts of interest. This does not alter our adherence to PLOS ONE policies on sharing data and materials, but please note that the anonymized data is only available upon request due to privacy reasons regarding patient's privacy and sensitive data.

**Abbreviations:** CI, Confidence Interval; DENV, Dengue Virus; ELISA, Enzyme-Linked Immunosorbent Assay; IgG, Immunoglobulin G; IgM, Immunoglobulin M; NS1, Non-Structural protein 1; RT-qPCR, Reverse-Transcriptase quantitative Polymerase Chain Reaction; SAE, Serious Adverse Event; SDC, Suspected Dengue Cases.

proportion of 1.4% (95%CI 0.5–3.8) was estimated. No severe cases or hospitalizations occurred during the study.

## Conclusion

Community-based active surveillance was shown as efficient to detect symptomatic dengue cases.

## Clinical trial registration

NCT02766088.

## Introduction

Dengue is a viral disease caused by four types of dengue viruses (DENV-1, DENV-2, DENV-3, and DENV-4) [1] that are transmitted by mosquito vectors, being primarily *Aedes aegypti* and secondarily *Aedes albopictus* [2, 3]. *Aedes aegypti* is present in the tropical and subtropical regions and well adapted to urban habitats [4]. *Aedes albopictus* has spread in North America and some Southern European countries as it can accommodate to cooler temperate climates [4]. Although most dengue infections are asymptomatic, clinical manifestations can range from febrile illness to potentially fatal dengue shock syndrome [2, 5, 6]. The course of the disease develops in up to three phases, starting with an acute febrile episode of three to seven days [5]. Fever can be followed by the critical phase characterized by a systemic vascular leak syndrome lasting for one or two days [5]. Most patients recover from the hemorrhagic episode, though the disease can progress to potentially fatal dengue shock syndrome, severe bleeding or organ failure [6].

Worldwide, the global incidence of dengue increased from approximately 500,000 reported cases in 2000 to more than three million in 2015, and the highest number of cases ever reported occurred in 2019 [7]. It was estimated that between 100 and 400 million dengue infections annually occur worldwide [8], and that half of the global population, living within 128 countries, is now at risk [9]. Dengue is endemic in Asia and also in the Western Pacific, the Americas, Africa, and the Eastern Mediterranean region [7]. Moreover, arbovirus vectors are known to spread in the United States and Europe due to urbanization, increased mobility, and climate change [10]. The increase in dengue incidence is accompanied by explosive outbreaks that are seasonal and influenced by characteristics of the vector and the host [6, 7]. Following the re-infestation of Latin Americas by the *Aedes aegypti* mosquito during the 1960s, several outbreaks of dengue were reported in the region [11]. In 1977, DENV-1 caused an epidemic that began in Jamaica that expanded to Mexico by 1978 [11]. In 1981, DENV-4 was introduced in the Caribbean and caused epidemics in several countries including Mexico where some cases of dengue hemorrhagic fever were observed [11]. During the 1990s, several epidemics were caused by the DENV-3 type in Mexico. Between 2000 and 2010, increased dispersion of *Aedes aegypti* has amplified dengue virus circulation, leading to several outbreaks, including in Mexico in 2009, where around 250,000 cases were reported [11]. In the Americas, the number of cases annually reported has increased from around 400,000 in 2000 to more than three million in 2019, with more than 25,000 cases classified as severe [12]. In Mexico, reported numbers of probable and confirmed cases increased between 2018 and 2019 from 78,621 to 268,458 and 12,706 to 41,505, respectively [13]. The annual cost associated with dengue in the Mexican population was estimated to be around 170 million US dollars, accompanied by an

annual average burden of 65 disability-adjusted life-years per million population [14]. In 2019, the overall rate for the country was 32.96 per 100,000 individuals, with the highest rate being in Jalisco (141.6/100,000) [13]. The increase in dengue incidence was particularly important in this state as estimates for 2016 and 2017 were 24.9 and 13.8 per 100,000 habitants, respectively [15, 16]. A large proportion of the increase was caused by DENV-2 in Mexico as the total number of DENV-2 cases progressed from 1,626 in 2017 to 12,637 in 2019 [13, 15].

Aside from the uncontrolled spreading of mosquito vectors, increased awareness by authorities and subsequent improvements in surveillance participate in the apparent increase in dengue infections. However, the lack of accurate incidence rates due to misdiagnosis and underreporting suggests that the burden of dengue might be even higher than the current estimates [7]. The underestimation of the dengue incidence arises from different factors: the high proportion of asymptomatic infections, the subjects with mild symptoms who do not seek treatment from a physician, the significant proportion of misdiagnoses due to the similarity of dengue symptoms and the underreporting of diagnosed cases [17, 18]. Compared to passive surveillance, active surveillance has the potential to detect mild and asymptomatic cases when coupled with serosurveillance and can help to understand the changes in epidemiology [17, 19]. Early detection of dengue would allow to improve the management of the disease and alleviate the overall burden of severe episodes and complications [20]. Moreover, estimates of unapparent, clinically apparent cases confirmed virologically would help assess the true disease burden and inform prevention strategies, including vaccine studies [21].

Currently, only one vaccine is available and licensed in about twenty countries [22]. Although this vaccine is efficacious and safe in seropositive individuals, a lower protection during the first two years, followed by an increased risk of severe dengue and hospitalization have been observed in seronegative vaccinated individuals [22]. Due to this risk of severe dengue in naïve individuals, vaccine use is limited to seropositive individuals. It is therefore important to achieve an understanding of disease dynamics in endemic areas to support further dengue vaccine development programs.

The present observational cohort study was conducted to primarily estimate the overall incidence of dengue infection confirmed by RT (reverse transcriptase)-PCR in subjects aged 6 months to 50 years living in a highly endemic area in Mexico.

Secondary objectives were to estimate a) the incidence of virologically (PCR or non-structural protein 1 [NS1]) confirmed and probable symptomatic dengue cases (by detection of IgG/IgM antibodies to dengue virus by ELISA or rapid immunochromatographic test) by age, gender, serotype (if applicable), and b) the prevalence of anti-DENV IgG antibodies at enrolment, overall and by age, as well as c) to describe the clinical presentations of dengue cases.

## Methods

### Study design and setting

This multi-center, prospective, household-based cohort surveillance study was conducted in geographically-defined communities in Latin America and Southeast Asia (NCT02766088). The initial protocol and study planning included seven countries. However, due to early study termination, only two sites in the Philippines and Mexico participated in the study. Here we report on the data collected during a 27-month period in Zapopan and neighboring municipalities in the state of Jalisco, Mexico, between July 14th, 2016 and September 14th, 2018.

### Ethical statement

This multi-center, prospective, household-based cohort surveillance study (NCT02766088) was conducted following the Internal Council on Harmonization good clinical practice

guidelines and the Declaration of Helsinki. The study protocol, amendments, and other study-related documents were reviewed and approved by the study site's independent ethics committee (research committee and research ethics committee of Hospital General de Occidente). Written and signed informed consent to participate were obtained from eligible individuals or their legally authorized representative. Additional assent from subjects below the legal age of consent was sought when applicable. The present manuscript was developed following the STrengthening the Reporting of OBservational studies in Epidemiology (STROBE) statement.

## Participants

Study researchers used a community-based approach for the recruitment of subjects. Invitation to the families was made through seven promoters of the study (three social workers, two doctors, a nutritionist, and a nurse) who also lived in Zapopan and the neighboring municipalities. Each promoter contacted three to eight families. The percentage of participants recruited by each promoter varied between 8% and 20%. Flyers with the information about the study and contact center were distributed. Some families were recruited through other families already included in the study. To be eligible for inclusion in the study, individuals should a) be 6 months to 50 years of age, b) live in Zapopan or neighboring municipalities (state of Jalisco, Mexico), c) agree to go to the study site for visit(s) in case of acute febrile illness, observe the signs of dengue and understand how to measure and report body temperature, d) plan to remain at the same residence during the study follow-up period, and e) be reachable by phone. Several members of the same household could participate in the study. Written and signed informed consent to participate were obtained from eligible individuals or their legally authorized representative. Additional assent from subjects below the legal age of consent was sought when applicable.

## Study procedures

A total of three visits were scheduled over a 24-months follow-up period (Fig 1). Socio-demographic information, medical and vaccination histories, as well as a blood sample, were collected at enrolment (Visit 1) through a questionnaire (S1 Appendix). The Month-12 and Month-24 visits occurred via telephone contact to collect serious adverse events (SAEs) and updated demographic, medical, and vaccination information. SAEs were defined as any death, life threatening event, hospitalization, or persistent disability that occurred during the study. SAEs related to blood drawn were considered as related to study procedure, other SAEs were also collected during the entire study period. Between the Month-12 and Month-24 visits, surveillance contacts for febrile illness were conducted once a week by telephone (95%) or in person (5%). Unscheduled visits at the study site were required in case of acute febrile illness to assess suspected dengue cases (SDC) (Fig 1). All SDCs were then followed in person.

**Determination of serostatus at study entry.** Anti-DENV (serotypes 1, 2, 3, and 4) immunoglobulin G (IgG) were measured by indirect enzyme-linked immunosorbent assay (ELISA) from the blood sample taken at enrolment (*Panbio Dengue IgG Indirect ELISA*) [23].

**Confirmation of dengue infection and serotype in suspected cases.** Confirmation and type of DENV infection were determined from the blood samples taken during unscheduled visits for SDCs. Assays for the determination of DENV types were systematically performed using the reverse-transcriptase quantitative polymerase chain reaction (RT-qPCR) technique (*Simplexa Dengue*, *Focus Diagnostics*) [24]. Viral RNA extracted from serum samples was reverse-transcribed into cDNA, thereafter detected by real-time PCR. The assay amplifies four serotype-specific regions (dengue 1: NS5 gene; dengue 2: NS3 gene; dengue 3: NS5 gene; dengue 4: capsid gene) that allow discriminating the serotypes.

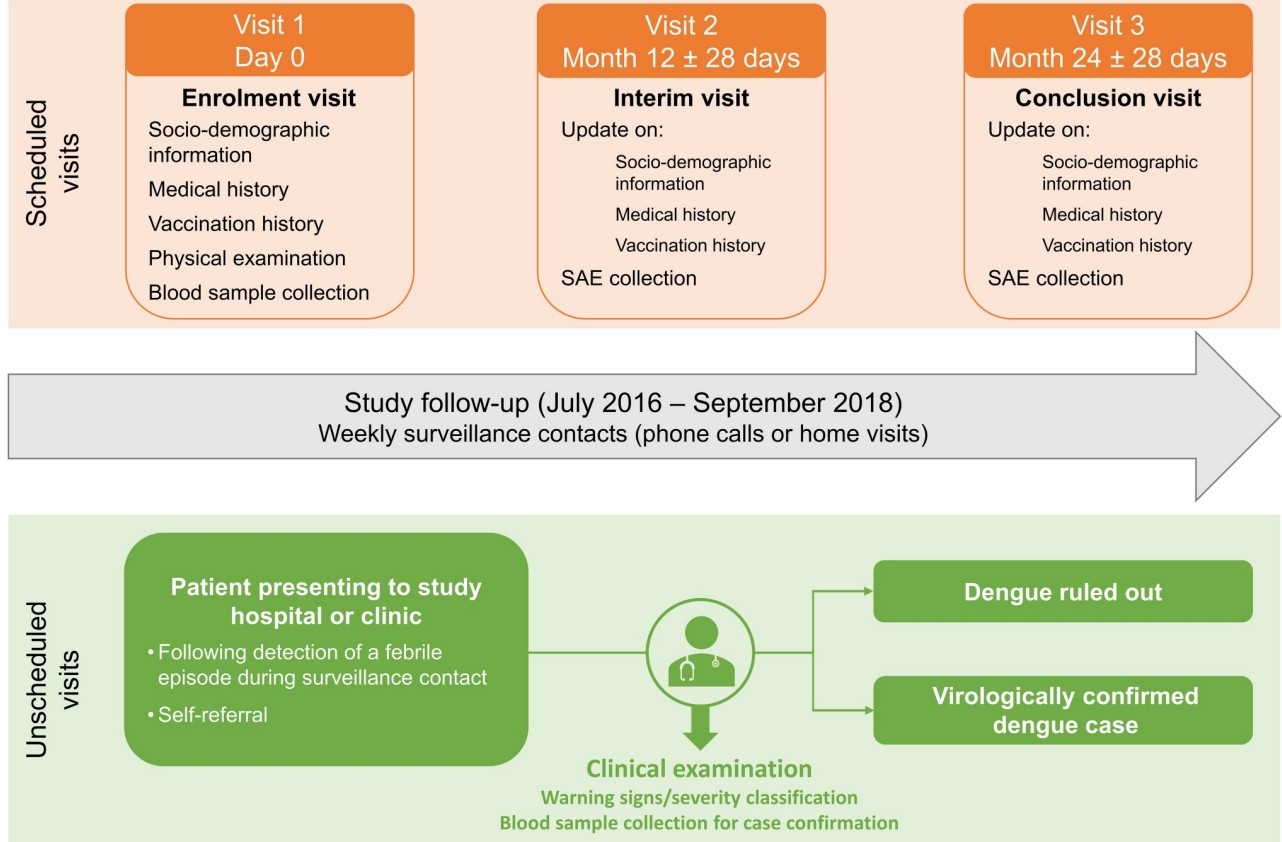

**Fig 1. Study design.** SAE, serious adverse event related to blood drawn. Virologically confirmed dengue case: confirmed by reverse-transcriptase quantitative polymerase chain reaction or detection of non-structural protein 1 antigen.

A rapid immunochromatographic test (*SD Bioline Dengue Duo*), consisting of a one-step assay designed to detect both dengue virus NS1 antigen and differential IgG/immunoglobulin M (IgM) antibodies to dengue virus, was primarily used to provide prompt laboratory results to the physician and the subject [25].

Due to the unavailability of the combined test, a one-step sandwich format microplate enzyme immunoassay (*Platelia Dengue NS1 Ag*) for the qualitative or semi-quantitative detection of dengue virus NS1 (NS1-antigen ELISA) [26] was performed in combination with IgG and IgM capture by ELISA (*Panbio*) [23].

## Case definitions

A *SDC* was considered in the presence of body temperature $\geq 38.0°C$ within the past eight days lasting from 36 hours to seven days, potentially accompanied by other dengue symptoms (see S2 Appendix). The association of symptoms with a potential dengue episode was based on physician's differential diagnosis.

A SDC presenting at the health care facility within five days following the symptom onset was defined as an early presenter. A SDC presenting at the health care facility six days or more after the onset of symptoms was defined as a late presenter.

Seven consecutive calendar days with body temperature $<38.0°C$, in the absence of antipyretic medication, were required to differentiate two episodes of suspected dengue [6].

A *virologically confirmed symptomatic dengue case* was an SDC confirmed by either RT-qPCR or NS1 antigen detection or both.

A *probable symptomatic dengue case* was defined as an SDC for which the RT-qPCR was either negative or not performed, had a negative or undetermined NS1 but (i) had a positive anti-DENV IgM text, or (ii) the anti-DENV IgG capture test was positive.

*Dengue cases with warning signs* had at least one aggravated symptom (see S2 Appendix).

*Severe dengue* had at least one severe symptom (see S2 Appendix).

## Statistical methods

Sample size calculation was performed for the whole study that should have included four study sites. Calculation assumed an annual dropout rate of 5% and targeted the enrolment of about 1,750 subjects across the four centers. In Zapopan, approximately 300 to 500 subjects were to be recruited. The target study population aimed at including between 30% and 50% of adults.

Statistical analysis was performed on all eligible participants with available data. Demographic characteristics (age at Visit 1, gender and the number of participants enrolled per household) were summarized using descriptive statistics. The incidence proportions were calculated during the study period for the RT-qPCR confirmed, the virologically confirmed, and the probable symptomatic dengue cases. These proportions were also estimated by generalized estimating equations logistic regression model accounting for the clustering effect (the households). The 95% confidence interval (CI) accounting for the clustering effect was computed for all estimated incidence proportions. However, if the estimated design effect was less than 1, then the classical logistic regression model not accounting for the clustering effect was used to estimate the incidence proportion and 95% CI (see S2 Appendix for the definition of design effect). The clinical symptoms reported during SDCs were tabulated. The proportion of subjects with a DENV antibody IgG positive result at enrolment and accompanying 95% CI were estimated by age category using the same methodology as for the incidence proportions. All statistical analyses were performed using the statistical analysis systems (SAS) version 9.4.

## Results

### Demographics

A total of 350 individuals from 87 households were enrolled, located mainly in the municipalities of Zapopan (69.4%) and Guadalajara (21.3%); other less frequent municipalities were Tlaquepaque, Tonalá, Tlajomulco, and Chapala (Jalisco, Mexico). A total of 344 subjects completed the study; five (1.4%) were lost to follow-up and one participant died of causes unrelated to the study. The number of subjects enrolled per household ranged from one to seven (median: 4; interquartile range [IQR]: 4–4). Of the 350 enrolled subjects, 184 (52.6%) were women, 188 (53.7%) were under 18 years of age and the remaining 162 (46.3%) participants were adults up to 50 years of age (Table 1).

### Serostatus at enrolment

Overall, 68 (19.4%) study participants were anti-DENV IgG positive at enrolment (Table 2). The highest anti-DENV IgG positivity rate (27.2%) was observed in the 18–50 years age group.

### Virological endpoints

A total of 28 unscheduled visits were reported from July 2016 to September 2018, among which 18 were assessed as SDCs and further evaluated. Among the ten other unscheduled

**Table 1. Demographics and households' characteristics.**

| Characteristics | Number of subjects, n (%) |
|---|---|
| Total number of enrolled subjects | 350 (100) |
| Gender | |
| Female | 184 (52.6) |
| Male | 166 (47.4) |
| Age groups | |
| 6–12 months | 2 (0.6) |
| 1–4 years | 47 (13.4) |
| 5–8 years | 43 (12.3) |
| 9–17 years | 96 (27.4) |
| 18–50 years | 162 (46.3) |
| Age at enrolment | |
| Mean (SD), years | 20.1 (13.99) |
| Median (IQR), years | 15.0 (8.0–32.0) |
| Min–Max | 9 months–50 years |
| | Number of households, n' (%) |
| Total number of enrolled households | 87 |
| Number of participants enrolled per households | |
| 1 | 1 (1.1) |
| 2 | 5 (5.7) |
| 3 | 12 (13.8) |
| 4 | 49 (56.3) |
| 5 | 15 (17.2) |
| 6 | 3 (3.4) |
| 7 | 2 (2.3) |

n, number of subjects; n', number of households; SD, standard deviation; IQR, interquartile range.

**Table 2. Proportion of subjects with anti-DENV IgG positive result at enrolment (ATP cohort).**

| | Number of subjects, n (%) | Proportions estimated from GEE[a], % (95%CI[b]) |
|---|---|---|
| Age groups | | |
| 6–12 months (N = 2) | 0 (0.0) | - |
| 1–4 years[c] (N = 47) | 3 (6.4) | 6.4 (2.1–18.0) |
| 5–8 years (N = 43) | 5 (11.6) | 11.8 (4.5–27.5) |
| 9–17 years (N = 95) | 16 (16.8) | 16.6 (9.8–26.6) |
| 18–50 years (N = 160) | 44 (27.5) | 27.2 (19.8–36.2) |
| Overall (N = 347) | 68 (19.6) | 19.4 (14.5–25.6) |

DENV, dengue virus; IgG, immunoglobulin G; N, total number of subject tested; n, number of subjects with positive results; ATP, according-to-protocol.

[a] Proportions estimated from generalised estimating equations (GEE) logistic regression model taking the clustering effect (the households) into account.

[b] 95%CI = 95% confidence interval based on the robust variance estimate from the GEE model.

[c] % = (n/N) X 100 and 95%CI = Wald CI as the design effect is $\leq$1.

visits, six subjects had upper respiratory tract infections, two had gastrointestinal infections, one had urinary tract infection, and the last subject had influenza and urinary tract infection. None of the SDCs were hospitalized. Among the 18 SDCs, one subject was IgG positive at enrollment. Sixteen subjects were early presenters, and two subjects were late presenters. Five suspected cases were virologically confirmed, either by both RT-qPCR and dengue NS1 antigen assay (n = 4) or only by dengue NS1 antigen assay (n = 1). All four RT-qPCR-confirmed cases were serotyped as DENV-1. The remaining 13 SDCs were tested negative by RT-qPCR for all 4 DENV types, by NS1, by DENV IgG, and by DENV IgM. Three of the five virologically confirmed symptomatic dengue cases occurred during the dengue season (July to October), the other two were reported in December 2016 and November 2017, outside the dengue season. Two cases concomitantly occurred in the same household.

The overall incidence proportion of RT-qPCR- and virologically confirmed symptomatic dengue cases over the 27-month study period was 1.1% (95% CI 0.4–3.0) and 1.4% (95% CI 0.5–3.8), respectively. No probable SDCs were reported.

## Clinical presentations

Among all five virologically confirmed cases, only one presented with at least one warning sign for dengue as defined by the World Health Organization [5]. Symptoms reported for the virologically confirmed symptomatic dengue cases and the other SDCs are presented in Table 3. All confirmed symptomatic dengue cases presented the following main signs of dengue: fever, headache, retroorbital pain, and myalgia. Three out of the five confirmed cases and ten out of the 13 non-confirmed SDCs had at least one digestive symptom. Two of the confirmed dengue cases and seven of the cases with no virological nor serological evidence of dengue infection had at least one respiratory symptom. Only one non-confirmed SDC had hemorrhagic manifestation (gingival bleeding and epistaxis). No severe cases were reported, and all patients recovered from their SDC with no complications.

## Safety outcomes

No SAEs related to study procedures were reported during the entire study period. One fatal outcome unrelated to the study procedures occurred during the study.

## Discussion

This study was conducted to evaluate the incidence of symptomatic dengue infections among household members aged 6 months to 50 years in Jalisco, the Mexican state having reported the highest dengue incidence (141.6/100,000) in 2019 [13]. The active weekly surveillance conducted from July 2016 to September 2018 detected 18 suspected dengue cases among which five were virologically confirmed.

The number of virologically confirmed dengue cases was close to the number expected at study design, yielding an incidence proportion of 1.4%. Moreover, the overall seroprevalence in this community (19.4%; 95% CI 14.5–25.6) was close to estimates from a cross-sectional study conducted in children aged 6–17 years old from 22 endemic states in Mexico in which the seroprevalence in seven clustered states including Jalisco was 13.3% (95% CI 9.0–19.2) [27]. This suggests that an important proportion of the population could be naïve to dengue, even in the adult population where the highest seroprevalence was observed. The trend of increased seroprevalence with age is also consistent with the results of the cross-sectional study [27]. Moreover, the serotyping results from the four RT-qPCR-confirmed cases were in accordance with the serotypes observed by the Mexican surveillance system of dengue as DENV-1 was almost exclusively detected in cases reported in Jalisco in 2016 and 2017 [15, 16]. However,

**Table 3. Summary of temperature at first visit and symptoms of suspected dengue cases.**

| Characteristics | Categories | Virologically confirmed N = 5 | | Non-confirmed SDC N = 13 | | Total N = 18 | |
|---|---|---|---|---|---|---|---|
| | | n | % | n | % | n | % |
| Axillary temperature at first visit [˚C] | <37.5 | 3 | 60.0 | 11 | 84.6 | 14 | 77.8 |
| | 37.5–38.0 | 0 | 0 | 0 | 0 | 0 | 0 |
| | 38.1–38.5 | 2 | 40.0 | 2 | 15.4 | 4 | 22.2 |
| | 38.6–39.0 | 0 | 0.0 | 0 | 0.0 | 0 | 0.0 |
| | >39.0 | 0 | 0.0 | 0 | 0.0 | 0 | 0.0 |
| At least one main sign | Yes | 5 | 100 | 13 | 100 | 18 | 100 |
| Fever | Yes | 5 | 100 | 13 | 100 | 18 | 100 |
| Headache | Yes | 5 | 100 | 13 | 100 | 18 | 100 |
| Retroorbital pain [eye pain] | Yes | 5 | 100 | 9 | 69.2 | 14 | 77.8 |
| Myalgia | Yes | 5 | 100 | 11 | 84.6 | 16 | 88.9 |
| Joint pain | Yes | 3 | 60.0 | 8 | 61.5 | 11 | 61.1 |
| Chills | Yes | 2 | 40.0 | 2 | 15.4 | 4 | 22.2 |
| Rash | Yes | 1 | 20.0 | 4 | 30.8 | 5 | 27.8 |
| Itching | Yes | 1 | 20.0 | 1 | 7.7 | 2 | 11.1 |
| At least one digestive symptom | Yes | 3 | 60.0 | 10 | 76.9 | 13 | 72.2 |
| Abdominal pain | Yes | 3 | 60.0 | 8 | 61.5 | 11 | 61.1 |
| Nausea or vomiting | Yes | 3 | 60.0 | 9 | 69.2 | 12 | 66.7 |
| Diarrhea | Yes | 2 | 40.0 | 2 | 15.4 | 4 | 22.2 |
| At least one respiratory symptom | Yes | 2 | 40.0 | 7 | 53.8 | 9 | 50.0 |
| Cough | Yes | 2 | 40.0 | 5 | 38.5 | 7 | 38.9 |
| Nasal Congestion | Yes | 2 | 40.0 | 4 | 30.8 | 6 | 33.3 |
| Sore throat | Yes | 0 | 0.0 | 2 | 15.4 | 2 | 11.1 |
| Dyspnea | Yes | 0 | 0.0 | 1 | 7.7 | 1 | 5.6 |
| At least one hemorrhagic manifestation | Yes | 0 | 0.0 | 1 | 7.7 | 1 | 5.6 |
| Petechiae | Yes | 0 | 0.0 | 0 | 0.0 | 0 | 0.0 |
| Purpura/ecchymosis | Yes | 0 | 0.0 | 0 | 0.0 | 0 | 0.0 |
| Hematemesis | Yes | 0 | 0.0 | 0 | 0.0 | 0 | 0.0 |
| Melena/hematochezia | Yes | 0 | 0.0 | 0 | 0.0 | 0 | 0.0 |
| Gingival bleeding | Yes | 0 | 0.0 | 1 | 7.7 | 1 | 5.6 |
| Epistaxis | Yes | 0 | 0.0 | 1 | 7.7 | 1 | 5.6 |
| Urinary tract bleeding | Yes | 0 | 0.0 | 0 | 0.0 | 0 | 0.0 |
| Unusual vaginal bleeding | Yes | 0 | 0.0 | 0 | 0.0 | 0 | 0.0 |
| At least one other sign | Yes | 3 | 60.0 | 8 | 61.5 | 11 | 61.1 |
| Pallor or cool skin | Yes | 0 | 0.0 | 0 | 0.0 | 0 | 0.0 |
| Conjunctivitis | Yes | 1 | 20.0 | 1 | 7.7 | 2 | 11.1 |
| Jaundice | Yes | 0 | 0.0 | 0 | 0.0 | 0 | 0.0 |
| Convulsion or coma | Yes | 0 | 0.0 | 0 | 0.0 | 0 | 0.0 |
| Lethargy or restlessness | Yes | 0 | 0.0 | 2 | 15.4 | 2 | 11.1 |
| Clinical fluid accumulation | Yes | 1 | 20.0 | 0 | 0.0 | 1 | 5.6 |
| Dizziness | Yes | 1 | 20.0 | 2 | 15.4 | 3 | 16.7 |
| Thoracic pain | Yes | 0 | 0.0 | 0 | 0.0 | 0 | 0.0 |
| Other | Yes | 3 | 60.0 | 5 | 38.5 | 8 | 44.4 |

N, total number of SDCs; n, number of SDCs with the symptom; SDC, suspected dengue case.

# Plain Language Summary

## What is the context?

- Dengue is a disease transmitted by mosquitos. While most dengue cases are asymptomatic, observed symptoms can vary from febrile illness like other diseases to death caused by severe bleeding or organ failure

- As a result, dengue cases are often misdiagnosed and underreported.

- Recent changes in dengue incidence and epidemiology have been reported in Zapopan municipality in Jalisco, one of the most affected Mexican states.

## What is new?

- We organized a weekly contact (either by phone or in person) with the study participants or their legal guardians (aged 6 months to 50 years) in Zapopan, Mexico, between July 2016 and September 2018. Participants showing acute febrile illness were sent to study hospital to confirm if they had contracted dengue.

- At study start, almost 20% of the participants were seropositive for dengue.

- Over the 27-month study period, the overall proportion of confirmed symptomatic dengue cases was between 1.1% (95% confidence interval [CI] 0.4 – 3.0) and 1.4% (95%CI 0.5 – 3.8) depending on the detection method used.

## What is the impact?

- This surveillance and screening approach helped to further identify the proportion of symptomatic infections.

- A high proportion of the population does not have dengue antibodies and is susceptible to the disease.

Fig 2. Plain language summary.

2019 reports have shown a dramatic increase in DENV-2 as well as in the incidence of dengue in Jalisco [13].

The findings presented here contribute to the body of evidence about the burden of dengue in Jalisco, an area where recent changes in incidence and epidemiology have been reported [13]. As for the assessment of influenza A and B burden in Mexico [28], our surveillance approach was effective in detecting symptomatic dengue cases and showed that an important proportion of the studied population was naïve to dengue. Moreover, this setting would be appropriate for the development of prophylactic dengue vaccines as better characterizing the serostatus of the population and the yearly incidence of symptomatic dengue cases are important pre-requisite when planning vaccine efficacy trials and estimating the sample size of these trials.

## Limitations

The recruitment process of participants by flyers provided by healthcare workers and the inclusion of multiple members from the same household constitute a selection bias. Moreover, the high proportion of weekly surveillance contacts made by telephone poses a risk of response bias. Another major limitation of our study is also related to the surveillance strategy, focusing on the acute febrile illness of at least two days of duration. The use of antipyretics or transient fever lasting less than two days may have hidden some potential dengue cases. Moreover, focusing on a clinical presentation that is common to many illnesses decreases the specificity of surveillance. As around three-quarters of dengue infections are asymptomatic, it is expected that overall dengue incidence was underestimated [5, 29]. An additional blood sample taken at study conclusion could have shed some light on the proportion of asymptomatic dengue infections that have occurred during the study conduct. Finally, atypical presentations of dengue have been described and may also have resulted in an underestimation of symptomatic cases [30, 31].

## Conclusion

Active surveillance was effective in detecting symptomatic dengue cases. However, screening programs are needed to further identify the proportion of asymptomatic viremic infections and describe their contribution to the global epidemiology and transmission of dengue in Mexico. Improved surveillance may help understanding the changes in clinical presentations of dengue infection and assessing more accurately the burden of dengue. Nevertheless, current detection of incident cases may be sufficient for the development of efficient prevention strategies. Fig 2 provides a plain language summary of the findings of this study.

## Supporting information

**S1 Appendix. Socio-demographic questionnaire.**
(PDF)

**S2 Appendix. Supplementary methods.** Cases definitions and sample size calculation.
(DOCX)

## Acknowledgments

The authors would like to thank the participants and their legal guardians for their participation to the study. The authors would also like to thank Jouda Aissa, Robert Paris, Veronique Bianco, Monica Garcia-Cuellar and Efriel Hazel Cruz for their significant contribution to the study. They also would like to thank the following members of the study staff: Maria del Carmen Lara-del Olmo, Guillermo Barboza-Alvarado, Monica D Ramirez-Castellanos and Lourdes A Lozano-Mercado. Finally, the authors thank the Business & Decision Life Sciences platform for editorial assistance and manuscript coordination, on behalf of GSK. Janne Tys coordinated the manuscript development and editorial support. Jonathan Ghesquiere provided medical writing support.

### Disclosures

#### Trademark statement

Simplexa is a trademark owned by or licensed to Focus Diagnostics. Panbio is a trademark owned by or licensed to Panbio Ltd. SD Bioline is a trademark owned by or licensed to the Abbott group of companies. Platelia is a trademark owned by or licensed to Bio-Rad.

## Author Contributions

**Conceptualization:** Rodrigo DeAntonio, Gerardo Amaya-Tapia, Gabriela Ibarra-Nieto, Gloria Huerta, Adrienne Guignard.

**Data curation:** Rodrigo DeAntonio, Gerardo Amaya-Tapia, Gabriela Ibarra-Nieto, Melanie de Boer.

**Formal analysis:** Rodrigo DeAntonio, Silvia Damaso, Melanie de Boer.

**Methodology:** Rodrigo DeAntonio, Gerardo Amaya-Tapia, Gabriela Ibarra-Nieto, Adrienne Guignard.

**Writing – review & editing:** Rodrigo DeAntonio, Gerardo Amaya-Tapia, Gabriela Ibarra-Nieto, Gloria Huerta, Silvia Damaso, Adrienne Guignard, Melanie de Boer.

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
