## [Decision Letter · Decision Letter 0]

18 Sep 2020

PONE-D-20-24564

Burden of dengue illness in Mexican people aged 6 months to 50 years old: results from a prospective, cohort study conducted in the Zapopan community

PLOS ONE

Dear Dr. de Boer,

Thank you for submitting your manuscript to PLOS ONE. After careful consideration, we feel that it has merit but does not fully meet PLOS ONE’s publication criteria as it currently stands. Therefore, we invite you to submit a revised version of the manuscript that addresses point by point raised by the reviewers. 

We look forward to receiving your revised manuscript.

Kind regards,

Nguyen Tien Huy, Ph.D., M.D.

Academic Editor

PLOS ONE

Journal Requirements:

3. Thank you for stating the following in the Financial Disclosure section :

'GlaxoSmithKline Biologicals SA funded this study (NCT02766088/GSK study identifier: 200318) and was involved in all stages of study conduct, including analysis of the data. GlaxoSmithKline Biologicals SA also covered all costs associated with the development and publication of this manuscript.'

We note that one or more of the authors have an affiliation to the commercial funders of this research study, GSK.

5. Please include captions for your Supporting Information files at the end of your manuscript, and update any in-text citations to match accordingly. Please see our Supporting Information guidelines for more information: http://journals.plos.org/plosone/s/supporting-information

Reviewers' comments:

Reviewer's Responses to Questions

**Comments to the Author**

1. Is the manuscript technically sound, and do the data support the conclusions?

Reviewer #1: Yes

Reviewer #2: Yes

Reviewer #3: Yes

2. Has the statistical analysis been performed appropriately and rigorously? 

Reviewer #1: Yes

Reviewer #2: I Don't Know

Reviewer #3: Yes

3. Have the authors made all data underlying the findings in their manuscript fully available?

Reviewer #1: Yes

Reviewer #2: Yes

Reviewer #3: No

4. Is the manuscript presented in an intelligible fashion and written in standard English?

Reviewer #1: Yes

Reviewer #2: Yes

Reviewer #3: Yes

5. Review Comments to the Author

Reviewer #1: First i would like to thank the authors for this really magnificent work conducted as overall it is well done

1/ the title should be changed from zapopan community - Jalisco state since we included other municipality other than zapopan

2/ I would appreciate if the authors mention some statistics about dengue in those <6 months worldwide or in Mexico then justify why that population was not included here

3/ For method please report the exact date of beginning and finishing the study and also precise exact duration was it 2 years of 27 months (some parts written 2 years other 27 months)

4/ The participants recruitment seems some kind of biased (selection bias) as they are people who go frequently to hospitals (contact with doctor nurse..) and are mostly from same households ( families recruited each other )

 Flyers process not elaborated , please precise the percentage of participants recruited by each promoter, by flyers, by other families , where no other procedure possible (radio, TV announcement maybe...)

- the selection bias should be mentioned in limitations

5/ The Socio-demographic information, medical and vaccination histories , physical exam - how were they done , any standardized method or guideline like a questionnaire maybe ? if so please provide a copy of it as supplementary material

6/ Weekly surveillance contact by telephone or in person are two different methods with very different results as in person is more reliable else there is risk of response bias

 give percentage of those followed by telephone and those by in person and how was that performed any standard questionnaire for that ?

- mention response bias in limitations of the study

7/ The unscheduled visits apart from the 18 suspected cases SDC , what were the reasons for the other unscheduled cases ?(10 visits)

8/ The death case what was the age and the cause of death ?

9/ The confirmed cases or suspected : were all of them followed by in person visits or by telephone ?

10/ Early presenter and late presenter , this classification was not used in the results or the table for fever , so when was the fever reported for the different cases ? when did they report to the hospital ? -- a timeline is appreciated

*else what are their clinical history (any chronic disease explaining their susceptibility to the disease) , their ages ?

11/ Were cases aware of the mosquito contact ? where did the contact happen ? the regions of the cases ?

12/ Serious adverse events SAE how were there defined ? any list of it ? how were they collected ? questionnaire ? physical exam ?

13/ Please include in the results that ''no severe cases' were reported'' and that ''all patients conditions did progress well with no complications''

14/ Please include in the method part a reference to STORBE checklist as justify the choice

Reviewer #2: I would like to give some suggestions for this paper.

-Throughout the paper: the paper should be consistent about the follow-up time (2 years or 27 months?). Also, there was no information on primary and secondary infections.

ABSTRACT - Conclusion: use a more specific word instead of “the surveillance approach.”

INTRODUCTION

-Paragraph 2: the paper should mention previous dengue outbreaks/epidemics.

METHODS

-The paper may mention the rationale of why the age range 6 months to 50 years was selected.

-The authors should describe how to identify suspected cases, such as mentioning the symptoms associated.

-Can the authors explain why the IgM test was not done?

RESULTS

-Clinical presentations: the authors should write more for this section. Many data from Table 1 should have been pointed out in this section, especially for the categories that had 0%.

DISCUSSION

-The first sentence of the Discussion should have the statistic for the incidence in 2019.

-Line 279 - 280: the paper should give the specific statistic (the incidence of DENV-1 compared to the others, how many percent increased of DENV-2 in 2019). Are there any reasons why the incidence dramatically increased in 2019?

-The paper should give a statistic about the population in Zapopan or neighboring municipalities (the living areas of participants) and discuss how their experimental results can represent the whole population.

-Line 286: I think the authors should explain more how their surveillance approach can help the development of vaccines.

-Are there any surveillance approaches prior to this study? If no, the authors should say that this is the first surveillance approach in the country and explain the significance of their study. If yes, the authors should compare their approach with previous strategies and explain the novelty/effectiveness of their approach.

TYPOS/GRAMMARS

-Write the full name of the test (such as reverse-transcriptase quantitative polymerase chain reaction) only the first time the word appears in the paper.

-Pay attention to format the name of genes (italic, lower-case for genes in the virus?)

-Some sentences need to be revised. The authors may need people who use English fluently to help improve the manuscript.

-The word “Though” at the beginning of the sentences should be replaced by “However”

Reviewer #3: The study is well conducted, fits the journal scope, and has a clear scientific message. However, I’d like to suggest these comments to increase the clarity of some points in the study:

1- Title: the authors in their paper described the disease incidence and not burden, which is completely different. I recommend replacing the word “Burden” with “Incidence” in the title. For more information about "Disease burden" concept, authors can search on WHO or CDC websites in addition to reading some articles discussing this concept.

2- Abstract: the sentence “The 13 remaining suspected cases were not confirmed” may mislead the readers that the study conductors didn’t perform RT-qPCR or NS1 assays for these 13 participants. So, it better to state clearly that their results were negative by these assays.

3- Introduction: it covers a good background for the topic, is well written, and the aim was described clearly. However, the section about vaccination [line 107 to line 115 on page 6 of 23] is clearly out of the topic and I think it should be removed.

4- Methods: it’s organized and clear. But additional clarification for these points should be added:

- Is there a justification for choosing this age range? I mean why not people older than 50.

- This sentence “Seven consecutive calendar days with body temperature <38.0°C, in the absence of antipyretic medication, were required to differentiate two episodes of SDC” should be supported with a reference or guideline.

- The first paragraph of statistical methods [line 203 to line 207 on pages 9 and 10 of 23] is confusing and should be re-written.

5- Results: in table 2, the results of serostatus were presented for only 247 out of 250 enrolled participants (the results of three participants were missed). As the blood samples were taken at the enrollment visit from all participants, the cause of missing for these three samples should be described in the results section.

6- ِAbout data sharing policy: you clearly stated that the data will be available for research proposals approved by an independent review committee if they submit proposals to (www.clinicalstudydatarequest.com) website, however, you didn't provide a clear ID/URL for your data in this website's database which is obligatory according to the journal policy.

6. PLOS authors have the option to publish the peer review history of their article (what does this mean?). If published, this will include your full peer review and any attached files.

Reviewer #1: **Yes: **Nacir Dhouibi

Reviewer #2: **Yes: **Anh Phuc Nguyet Nguyen

Reviewer #3: No

---

## [Author Response · Author response to Decision Letter 0]

1 Dec 2020

Dear Editor,

We would like to thank you for the consideration of our manuscript entitled “Burden of dengue illness in Mexican people aged 6 months to 50 years old: results from a prospective, cohort study conducted in the Zapopan community” (PONE-D-20-24564). We also would like to extend our thanks to the reviewers for their careful review of the manuscript.

We have addressed all comments from the reviewers below and revised the manuscript using track-changes. The line numbers used in the responses refers to the line numbers in the track-changes version. We believe that these revisions have helped us to improve our manuscript.

Additionally, please find hereafter the amended disclosure sections:

Funding statement:

GlaxoSmithKline Biologicals SA funded this study (NCT02766088/GSK study identifier: 200318). GlaxoSmithKline Biologicals SA also provided support in the form of salaries for authors GH, SD, MDB and AG. GlaxoSmithKline Biologicals SA was involved in all stages of study conduct, including study design, data collection and analysis; GlaxoSmithKline Biologicals SA also covered all costs associated with the development and publication of this manuscript.

Competing Interests Statement:

GlaxoSmithKline Biologicals SA (GSK) funded this study and covered all costs associated with the development and publication of this manuscript. GH, SD, MDB and AG are employees of the GSK group of companies. SD and AG hold shares in the GSK group of companies. RD was an employee of the GSK group of companies at the time of the study. GAT received payments from the GSK group of companies, as part of the multi-center study. GH, SD, MDB, AG and GAT declare no other financial and non-financial relationships and activities. GIN declare no financial and non-financial relationships and activities and no conflicts of interest.

This does not alter our adherence to PLOS ONE policies on sharing data and materials, but please note that the anonymized data is only available upon request due to privacy reasons regarding patient’s privacy and sensitive data.

We look forward to future correspondence regarding our submission and are more than happy to provide further information on any questions or comments you may have.

With kind regards,

Melanie de Boer

GSK

Email melanie.x.de-boer@gsk.com

 

Journal's comment

"1. Please ensure that your manuscript meets PLOS ONE's style requirements, including those for file naming. The PLOS ONE style templates can be found at:

https://journals.plos.org/plosone/s/file?id=ba62/PLOSOne_formatting_sample_title_authors_affiliations.pdf"

Response: We have revised the main text, placement of tables and figure captions as well as the title page.

"2. We note that you have indicated that data from this study are available upon request. PLOS only allows data to be available upon request if there are legal or ethical restrictions on sharing data publicly. For information on unacceptable data access restrictions, please see http://journals.plos.org/plosone/s/data-availability#loc-unacceptable-data-access-restrictions.

We will update your Data Availability statement on your behalf to reflect the information you provide."

Response: As explained in the previously provided data sharing statement, the data set is only available upon request because of privacy reasons regarding study participants. Indeed, the data contain potentially identifying or sensitive patient information. The anonymized data is available to anyone upon request on https://www.clinicalstudydatarequest.com/ (instructions on how to request the data is available directly on the link). The study identifier is 200318.

3. We note that one or more of the authors have an affiliation to the commercial funders of this research study, GSK.

Within your Competing Interests Statement, please confirm that this commercial affiliation does not alter your adherence to all PLOS ONE policies on sharing data and materials by including the following statement: ""This does not alter our adherence to PLOS ONE policies on sharing data and materials.” (as detailed online in our guide for authors http://journals.plos.org/plosone/s/competing-interests). If this adherence statement is not accurate and there are restrictions on sharing of data and/or materials, please state these. Please note that we cannot proceed with consideration of your article until this information has been declared.

Response:

a. Amended funding statement: “

GlaxoSmithKline Biologicals SA funded this study (NCT02766088/GSK study identifier: 200318). GlaxoSmithKline Biologicals SA also provided support in the form of salaries for authors GH, SD, MDB and AG. GlaxoSmithKline Biologicals SA was involved in all stages of study conduct, including study design, data collection and analysis; GlaxoSmithKline Biologicals SA also covered all costs associated with the development and publication of this manuscript.”

The role of the authors as mentioned in the ‘author contributions’ section is correct and accurately depicts the role of each author in the study.

“RD, GIN and AG participated to the conception and design of the study. GAT, RD, GIN and MDB contributed to the acquisition of data. GAT, RD, MDB and SD participated to the analysis and interpretation of data. All authors revised the article critically for important intellectual content and provided final approval of the submitted version.”

b. The competing Interests Statement already discloses the financial relationships of the authors with the funder. We have amended it to include the funder’s role and the adherence statement with justification on data access restriction.

“GlaxoSmithKline Biologicals SA (GSK) funded this study and covered all costs associated with the development and publication of this manuscript. GH, SD, MDB and AG are employees of the GSK group of companies. SD and AG hold shares in the GSK group of companies. RD was an employee of the GSK group of companies at the time of the study. GAT received payments from the GSK group of companies, as part of the multi-center study. GH, SD, MDB, AG and GAT declare no other financial and non-financial relationships and activities. GIN declare no financial and non-financial relationships and activities and no conflicts of interest.

This does not alter our adherence to PLOS ONE policies on sharing data and materials, but please note that the anonymized data is only available upon request due to privacy reasons regarding patient’s privacy and sensitive data.”

Response: Both abstracts should now be identical.

5. Please include captions for your Supporting Information files at the end of your manuscript, and update any in-text citations to match accordingly. Please see our Supporting Information guidelines for more information: http://journals.plos.org/plosone/s/supporting-information

Response: We have added captions for the Supplementary Appendix 1 and 2 (lines 555-557) and revised the supplementary files accordingly.

Response: We have moved the ethics statement in the method section (lines 152-165).

Reviewer #1

1/ the title should be changed from zapopan community - Jalisco state since we included other municipality other than zapopan 

Response: We have revised the title to “Incidence of dengue illness in Mexican people aged 6 months to 50 years old: results from a prospective, cohort study conducted in the Jalisco state”

2/ I would appreciate if the authors mention some statistics about dengue in those <6 months worldwide or in Mexico then justify why that population was not included here

Response: This epidemiological study aimed at enrolling a population in the age range that would be considered subsequently for an efficacy trial of a candidate vaccine. It is unlikely that infants younger than 6-month old would be enrolled in such trials for the reasons explained below.

In dengue endemic countries, most infants have dengue maternal antibodies at birth. The titers progressively decline in the first months of life. Maternally acquired DENV-specific antibodies play a dual role in infants during the first year of life: they confer protection at birth, and then they decline to a lower level capable of increasing the risk of severe DENV infection through antibody-dependent enhancement.

In addition, maternal antibodies may interfere with the response to primary infant vaccination (as observed for instance for measles or pertussis). There seems to be little opportunity for an immunogenic vaccine response among infants before 6 months of age.

We did not amend the manuscript with this rationale as we were asked to keep the text about vaccination to a minimum.

3/ For method please report the exact date of beginning and finishing the study and also precise exact duration was it 2 years of 27 months (some parts written 2 years other 27 months)

Response: We have added clarifications about the 27-month study period and harmonize throughout the text. Please note that the study lasted 27 months but that the follow-up period for each enrolled subject was planned for 24 months. Exact date of study start and conclusion were added in the methods (lines 144-146).

4/ The participants recruitment seems some kind of biased (selection bias) as they are people who go frequently to hospitals (contact with doctor nurse..) and are mostly from same households ( families recruited each other )

 Flyers process not elaborated , please precise the percentage of participants recruited by each promoter, by flyers, by other families , where no other procedure possible (radio, TV announcement maybe...) 

- the selection bias should be mentioned in limitations"

Response: We have added the number families contacted by the promoters and the proportion of patients recruited by each promoter (lines 170-171). We have added the selection bias in the limitation section (lines 364-366).

5/The Socio-demographic information, medical and vaccination histories, physical exam - how were they done, any standardized method or guideline like a questionnaire maybe? if so please provide a copy of it as supplementary material.

Response: Socio-demographic information were collected through a questionnaire in Spanish. We have added the information in the methods (lines 186-187) and provided the questionnaire as supplementary appendix 1.

6/ Weekly surveillance contact by telephone or in person are two different methods with very different results as in person is more reliable else there is risk of response bias

 give percentage of those followed by telephone and those by in person and how was that performed any standard questionnaire for that?

- mention response bias in limitations of the study "

Response: We have included the percentage of telephone contact (95%) and in person visits (5%) in bracket in the methods (line 194) and added a sentence about the risk of response bias (lines 366-367).

7/ The unscheduled visits apart from the 18 suspected cases SDC, what were the reasons for the other unscheduled cases? (10 visits) 

Response: We have added as sentence in the "Virological endpoints" section (lines 299-301)

8/ The death case what was the age and the cause of death? 

Response: The subject who died belonged to the 9-17 years group. As the death was not related to the study, this information and the cause of death are not provided to the reader.

9/ The confirmed cases or suspected: were all of them followed by in person visits or by telephone ?

Response: They were all followed in person. We have added lines 199-197.

10/ Early presenter and late presenter, this classification was not used in the results or the table for fever , so when was the fever reported for the different cases ? when did they report to the hospital ? -- a timeline is appreciated

*else what are their clinical history (any chronic disease explaining their susceptibility to the disease) , their ages ? "

Response: We have added the numbers of early and late presenters (lines 303-304). However, we did not collect clinical history or chronic conditions for SDCs.

11/ Were cases aware of the mosquito contact? where did the contact happen? the regions of the cases ? 

Response: Questions about mosquito contact were not asked to the subjects attending an ad-hoc visit. Regions of the cases is related to the study site location and enrollement criteria, i.e. Zapopan or neighboring municipalities (Jalisco state, Mexico), as indicated in the methods.

12/ Serious adverse events SAE how were there defined? any list of it? how were they collected? questionnaire? physical exam? 

Response: Only SAEs following blood drawn were considered as related to the study procedure. Definition of all SAEs was added to the methods (lines 189-192).

13/ Please include in the results that ''no severe cases' were reported'' and that ''all patients conditions did progress well with no complications''

Response: We have added the recommended text to the "Clinical presentation" section (lines 325-326).

14/ Please include in the method part a reference to STORBE checklist as justify the choice

Response: We have added the reference to the STROBE checklist in the ethical statement that now lies in the Methods (lines 163-165).

Reviewer #2

Throughout the paper: the paper should be consistent about the follow-up time (2 years or 27 months?). Also, there was no information on primary and secondary infections.

Response: The 27-month period is for the study duration. Follow-up of each subject was 24 months. We have clarified this in the methods (lines 144, 184). We have also added the information about IgG status of the SDCs in the "Virological endpoints" section (lines 302-303).

ABSTRACT - Conclusion: use a more specific word instead of “the surveillance approach .”

Response: Changed to "Community-based active surveillance"

"INTRODUCTION

-Paragraph 2: the paper should mention previous dengue outbreaks/epidemics. "

Response: We have added a brief history of dengue epidemics in the Americas, focusing on Mexico (lines 80-88).

"METHODS

-The paper may mention the rationale of why the age range 6 months to 50 years was selected . "

Response: This epidemiological study aimed at enrolling a population in the age range that could be subsequently considered for an efficacy trial of a candidate vaccine. In dengue endemic countries, most infants have dengue maternal antibodies at birth, that protect them from early dengue infection. However, maternal antibody levels wane over few months. Children younger than 6-month old are thus unlikely to be considered for vaccination. In addition, maternal antibodies may interfere with the response to primary infant vaccination (as observed for instance for measles or pertussis). 

The upper age limit is also related to the potential vaccine trial, limiting our assessment to age groups in which most individuals are healthy and immunocompetent. We broadly targeted adolescents and young adults as the incidence of dengue was shown to peak in the age range of 10 to 20 years in Mexico.

We did not amend the manuscript with this rationale as we were asked to keep the text about vaccination to a minimum.

The authors should describe how to identify suspected cases, such as mentioning the symptoms associated. 

Response: This is already presented in the first paragraph of the case definitions (lines 224-228) and complemented by Supplementary Appendix 2.

Can the authors explain why the IgM test was not done? 

Response: IgG/IgM test was done and used to assess suspected dengue cases. We have amended lines 132-135 to clarify this.

"RESULTS

-Clinical presentations: the authors should write more for this section. Many data from Table 1 should have been pointed out in this section, especially for the categories that had 0%. "

Response: We have added few sentences about the main observations from table 3 in the "clinical presentation" section. However, we believe that describing the null categories has less interest for the reader.

"DISCUSSION

-The first sentence of the Discussion should have the statistic for the incidence in 2019 ."

Response: We have added the values presented in the introduction.

Line 279 - 280: the paper should give the specific statistic (the incidence of DENV-1 compared to the others, how many percent increased of DENV-2 in 2019). Are there any reasons why the incidence dramatically increased in 2019? 

Response: As our study ended in September 2018, we would prefer to not discuss 2019 data in the results. The increased incidence is mentioned in the introduction as background information about the study location. We can however mention that the incidence of dengue globally is thought to increase in the future, most likely as a collateral effect of global warming. We have nevertheless added a sentence about the impressive increase of DENV-2 cases between 2017 and 2019 (lines 98-100) in Mexico.

The paper should give a statistic about the population in Zapopan or neighboring municipalities (the living areas of participants) and discuss how their experimental results can represent the whole population.

Response: Statistics about dengue in Jalisco and Mexico are provided in the discussion. As we did not calculate incidence, our observations could not be discussed in the light of the available data for dengue incidence in the state of Jalisco. We nevertheless believe that our observations are representative of the population living in Jalisco.

Line 286: I think the authors should explain more how their surveillance approach can help the development of vaccines. 

Response: We have completed the sentence to clarify the statement (lines 360-363).

Are there any surveillance approaches prior to this study? If no, the authors should say that this is the first surveillance approach in the country and explain the significance of their study. If yes, the authors should compare their approach with previous strategies and explain the novelty/effectiveness of their approach. 

Response: Such surveillance approach was already used for the assessment of influenza A and B co-circulation and burden in Mexico. We have added this information and the reference (28) in lines 356-357.

"TYPOS/GRAMMARS

-Write the full name of the test (such as reverse-transcriptase quantitative polymerase chain reaction) only the first time the word appears in the paper.

-Pay attention to format the name of genes (italic, lower-case for genes in the virus?)

-Some sentences need to be revised. The authors may need people who use English fluently to help improve the manuscript.

-The word “Though” at the beginning of the sentences should be replaced by “However”

Response: We have revised the text and corrected it where needed. 

Reviewer #3

1- Title: the authors in their paper described the disease incidence and not burden, which is completely different. I recommend replacing the word “Burden” with “Incidence” in the title. For more information about "Disease burden" concept, authors can search on WHO or CDC websites in addition to reading some articles discussing this concept.

Response: We have revised the title to “Incidence of dengue illness in Mexican people aged 6 months to 50 years old: results from a prospective, cohort study conducted in the Jalisco state”

2- Abstract: the sentence “The 13 remaining suspected cases were not confirmed” may mislead the readers that the study conductors didn’t perform RT-qPCR or NS1 assays for these 13 participants. So, it better to state clearly that their results were negative by these assays.

Response: We have corrected the sentence as suggested by the reviewer.

3- Introduction: it covers a good background for the topic, is well written, and the aim was described clearly. However, the section about vaccination [line 107 to line 115 on page 6 of 23] is clearly out of the topic and I think it should be removed.

Response: We have reduced the paragraph but kept some information about the vaccine as the present study aims at supporting the design of future efficacy study for novel vaccines. 

"4- Methods: it’s organized and clear. But additional clarification for these points should be added:

- Is there a justification for choosing this age range? I mean why not people older than 50. "

Response: This epidemiological study aimed at enrolling a population in the age range that could be subsequently considered for an efficacy trial of a candidate vaccine. In dengue endemic countries, most infants have dengue maternal antibodies at birth, that protect them from early dengue infection. However, maternal antibody levels wane over few months. Children younger than 6-month old are thus unlikely to be considered for vaccination. In addition, maternal antibodies may interfere with the response to primary infant vaccination (as observed for instance for measles or pertussis). 

The upper age limit is also related to the potential vaccine trial, limiting our assessment to age groups in which most individuals are healthy and immunocompetent. We broadly targeted adolescents and young adults as the incidence of dengue was shown to peak in the age range of 10 to 20 years in Mexico.

We did not amend the manuscript with this rationale as we were asked to keep the text about vaccination to a minimum.

This sentence “Seven consecutive calendar days with body temperature <38.0°C, in the absence of antipyretic medication, were required to differentiate two episodes of SDC” should be supported with a reference or guideline.

Response: Study protocol was developed following the WHO guidelines for diagnosis of dengue. We have repeated the ref 6 in line 234.

The first paragraph of statistical methods [line 203 to line 207 on pages 9 and 10 of 23] is confusing and should be re-written.

Response: We have revised the sentence for clarity.

5- Results: in table 2, the results of serostatus were presented for only 247 out of 250 enrolled participants (the results of three participants were missed). As the blood samples were taken at the enrollment visit from all participants, the cause of missing for these three samples should be described in the results section. 

Response: There were 350 subjects enrolled in Mexico, among whom 68 were positive, 279 were negative, and 3 had equivocal results. Equivocal results mean that the level of measured dengue IgG antibodies were not high enough to conclude of a dengue infection or not. The 3 subjects with equivocal lab results were excluded from the stat analysis in the ATP cohort.

6- ِAbout data sharing policy: you clearly stated that the data will be available for research proposals approved by an independent review committee if they submit proposals to (www.clinicalstudydatarequest.com) website, however, you didn't provide a clear ID/URL for your data in this website's database which is obligatory according to the journal policy.

Response: 

We have added details and explanation on data request in the response letter as requested by the journal. The URL provided leads directly to the instructions on how to request a data set and the study ID is 200318.

---

## [Decision Letter · Decision Letter 1]

3 Mar 2021

PONE-D-20-24564R1

Incidence of dengue illness in Mexican people aged 6 months to 50 years old: results from a prospective, cohort study conducted in the Jalisco state

PLOS ONE

Dear Dr. de Boer,

Thank you for submitting your manuscript to PLOS ONE. After careful consideration, we feel that it has merit but does not fully meet PLOS ONE’s publication criteria as it currently stands. Therefore, we invite you to submit a revised version of the manuscript that addresses the points raised during the review process.

PPlease modify the manuscript with the grammar suggestions provided by the reviewers.

We look forward to receiving your revised manuscript.

Kind regards,

Humberto Lanz-Mendoza

Academic Editor

PLOS ONE

Journal Requirements:

Reviewers' comments:

Reviewer's Responses to Questions

**Comments to the Author**

1. If the authors have adequately addressed your comments raised in a previous round of review and you feel that this manuscript is now acceptable for publication, you may indicate that here to bypass the “Comments to the Author” section, enter your conflict of interest statement in the “Confidential to Editor” section, and submit your "Accept" recommendation.

Reviewer #1: All comments have been addressed

Reviewer #2: All comments have been addressed

Reviewer #3: All comments have been addressed

Reviewer #4: All comments have been addressed

2. Is the manuscript technically sound, and do the data support the conclusions?

Reviewer #1: Yes

Reviewer #2: Yes

Reviewer #3: Yes

Reviewer #4: Yes

3. Has the statistical analysis been performed appropriately and rigorously? 

Reviewer #1: I Don't Know

Reviewer #2: I Don't Know

Reviewer #3: Yes

Reviewer #4: Yes

4. Have the authors made all data underlying the findings in their manuscript fully available?

Reviewer #1: No

Reviewer #2: Yes

Reviewer #3: No

Reviewer #4: Yes

5. Is the manuscript presented in an intelligible fashion and written in standard English?

Reviewer #1: Yes

Reviewer #2: Yes

Reviewer #3: Yes

Reviewer #4: Yes

6. Review Comments to the Author

Reviewer #1: (No Response)

Reviewer #2: Below are some suggestions about grammars to help improve this manuscript

-Throughout the paper, the author should write either “the state of Jalisco” or simply “Jalisco” instead of “Jalisco state”

-Suggested title: “Incidence of dengue illness in Mexican people aged 6 months to 50 years old: a prospective cohort study conducted in Jalisco”

-Line 41: no space between the number 19.4 and %

-Line 46: “cases were tested negative by these assay”

-Line 49: delete the word “conduct”

-Line 70: “Worldwide, the global incidence of dengue”

-Line 85-88: these three sentences are all simple sentences and should be reworded and/or combined together

-Line 112-113: the phrase “using serotype-specific polymerase chain reaction (PCR) and anti-dengue immunoglobulin G (DENV IgG) assays” is not relevant to the previous clause in the sentence. The author may consider deleting that phrase.

-Line 124: replace the word “vaccinees” by “vaccinated people” or “vaccinated individuals”

-Line 146: 14th, 2016 and September 14th, 2018.

-Line 177: replace the word “take” by “measure”

-Line 219: the author may consider changing the phrase “As an alternative to the combined test was not available” to “Due to the unavailability of the combined test”

-Line 270: no comma between “(69.4%)” and “and Guadalajara”

-Line 271: should have a comma between “Tlajomulco” and “and Chapala”

-Line 272: should not begin a sentence by a number

-Line 355: “dengue in Jalisco, in an area where”

-Line 364: the author can consider adding a subheader called “Limitations”

-Line 380: change the word “successful” to “effective”

Reviewer #3: (No Response)

Reviewer #4: There is an issue that the authors need to clarify between lines 135 abd 147. Other than that, my comments are minor.

7. PLOS authors have the option to publish the peer review history of their article (what does this mean?). If published, this will include your full peer review and any attached files.

Reviewer #1: **Yes: **Nacir Dhouibi

Reviewer #2: **Yes: **Anh Phuc Nguyet Nguyen

Reviewer #3: No

Reviewer #4: No

---

## [Author Response · Author response to Decision Letter 1]

30 Mar 2021

March 30, 2021

Dear Editor,

We would like to thank you for the consideration of our revised manuscript entitled “Burden of dengue illness in Mexican people aged 6 months to 50 years old: results from a prospective, cohort study conducted in the Zapopan community” (PONE-D-20-24564R1).

We have addressed all additional comments from the reviewers below and revised the manuscript using track-changes. We hope that the revised manuscript will fully meet PLOS ONE’s publication criteria.

We look forward to future correspondence regarding our submission and are more than happy to provide further information on any questions or comments you may have.

With kind regards,

Melanie de Boer 

GSK

Email melanie.x.de-boer@gsk.com

 

Comments: 

Reviewer #2

-Throughout the paper, the author should write either “the state of Jalisco” or simply “Jalisco” instead of “Jalisco state”

Reply: We made the correction throughout the text

-Suggested title: “Incidence of dengue illness in Mexican people aged 6 months to 50 years old: a prospective cohort study conducted in Jalisco”

Reply: We have modified the title as suggested.

-Line 41: no space between the number 19.4 and %

Reply: The space between numerals and the % has been removed (line 38 of the revised version with tracked changes).

-Line 46: “cases were tested negative by these assay”

Reply: We have revised the sentence as suggested (line 43 of the revised version with tracked changes).

-Line 49: delete the word “conduct”

Reply: Deleted (line 46 of the revised version with tracked changes)

-Line 70: “Worldwide, the global incidence of dengue”

Reply: We have inserted “global” in the sentence (line 67 of the revised version with tracked changes).

-Line 85-88: these three sentences are all simple sentences and should be reworded and/or combined together

Reply: We have revised the sentences to improve the flow (lines 81-87 of the revised version with tracked changes).

-Line 112-113: the phrase “using serotype-specific polymerase chain reaction (PCR) and anti-dengue immunoglobulin G (DENV IgG) assays” is not relevant to the previous clause in the sentence. The author may consider deleting that phrase.

Reply: We have deleted that part of the sentence (lines 111-112 of the revised version with tracked changes).

-Line 124: replace the word “vaccinees” by “vaccinated people” or “vaccinated individuals”

Reply: “Vaccinees” has been changed for “vaccinated individuals” (line 120 of the revised version with tracked changes).

-Line 146: 14th, 2016 and September 14th, 2018.

Reply: Corrected (lines 141-142 of the revised version with tracked changes).

-Line 177: replace the word “take” by “measure”

Reply: Replaced as suggested (line 168 of the revised version with tracked changes).

-Line 219: the author may consider changing the phrase “As an alternative to the combined test was not available” to “Due to the unavailability of the combined test”

Reply: Revised as suggested (line 208 of the revised version with tracked changes).

-Line 270: no comma between “(69.4%)” and “and Guadalajara”

Reply: Corrected (line 257 of the revised version with tracked changes).

-Line 271: should have a comma between “Tlajomulco” and “and Chapala”

Reply: Corrected (line 38 of the revised version with tracked changes).

-Line 272: should not begin a sentence by a number

Reply: Corrected (line 259 of the revised version with tracked changes).

-Line 355: “dengue in Jalisco, in an area where”

Reply: We have removed “in” before “an area where” (line 342 of the revised version with tracked changes).

-Line 364: the author can consider adding a subheader called “Limitations”

Reply: We have added the subheading for limitations (line 350 of the revised version with tracked changes).

-Line 380: change the word “successful” to “effective”

Reply: Revised as suggested (line 366 of the revised version with tracked changes).

Reviewer #4

Methods

LINES 135-136

However, due to early study termination, only two sites in the Philippines and Mexico participated in the study.

LINES 146-47

However, due to early study termination, only two sites in the Philippines and Mexico participated in the study.

Mention this sentence just once in the paper.

Reply: We have deleted the sentence from lines 146-47 (lines 149-151 of the revised version with tracked changes)

If the Philippines participated, where are the data and the analyses? Please clarify 

Expound briefly on “early study termination”

Reply: As the global, pre-defined analysis could not be performed, data from the Philippines and Mexico were analyzed separately. Data and results for the Philippines are described in another paper currently under consideration for publication in Asian Biomedicine.

In December 2017, GSK notified its longstanding collaboration partners, Institution Oswaldo Cruz (Fiocruz) and the Walter Reed Army Institute of Research and the US Army Medical Materiel Development Agency (WRAIR/USAMMDA), of the intention of GSK’s Vaccines Investment Board (VIB) to deprioritize the development of the dengue purified inactivated vaccine (DPIV) candidate. This decision was made due to the scientific challenges and development risks associated with the candidate vaccine. This decision entailed the early termination of the associated observational studies sponsored by GSK.

Page 16 Top row

Conjunctivis (spelling)

Reply: Corrected to “conjunctivitis”, thank you!

Page 12 Table 1

Age groups: The authors listed 5 levels, 1-4 and 5-8 years could be combined as they did not differ much in values and that the range would be closer to 9-17 and 18-50 years

Reply: We acknowledge that 1-4 and 5-8 age groups do not differ much in values and could have been combined. However, the stratification was made as per protocol and merging both group would require an additional analysis that would take time while having little impact on our observations and the overall message of the paper. 

LINE 256 household ranged from one to seven (median: 4).

The authors reported IQR in Table 1 but not in LINE 256

Reply: IQR is provided is Table 1 for the median age at enrolment. We have nevertheless added the IQR in text for the number of participants by household (lines 261-262 of the revised version with tracked changes).

LINES 247-248 incidence proportions. All statistical analyses were performed using the statistical analysis systems version 9.4. Cite the source of the system used

Reply: We have added the software, SAS, in brackets in the sentence (line 253 of the revised version with tracked changes).

LINE 233 Statistical analysis was performed on all evaluable eligible participants 

What is evaluable?

Reply: We have removed evaluable and inserted “with complete data” instead. We hope that this revision clarifies the meaning of the sentence (lines 238-239 of the revised version with tracked changes).

---

## [Editor Report · Decision Letter 2]

5 Apr 2021

Incidence of dengue illness in Mexican people aged 6 months to 50 years old: a prospective cohort study conducted in Jalisco

PONE-D-20-24564R2

Dear Dr. de Boer,

We’re pleased to inform you that your manuscript has been judged scientifically suitable for publication and will be formally accepted for publication once it meets all outstanding technical requirements.

Kind regards,

Humberto Lanz-Mendoza

Academic Editor

PLOS ONE
---

## [Editor Report · Acceptance letter]

22 Apr 2021

PONE-D-20-24564R2 

Incidence of dengue illness in Mexican people aged 6 months to 50 years old: a prospective cohort study conducted in Jalisco 

Dear Dr. de Boer:

I'm pleased to inform you that your manuscript has been deemed suitable for publication in PLOS ONE. Congratulations! Your manuscript is now with our production department. 

Kind regards, 

on behalf of

Dr. Humberto Lanz-Mendoza 

Academic Editor

PLOS ONE